# Dynamic natural components and morphological changes in nonculprit subclinical atherosclerosis in patients with acute coronary syndrome and mild chronic kidney disease at the 1-year follow-up and clinical significance at the 5-year follow-up

**Jia-cong Nong[1,2ʘ], Wei You[1ʘ], Yi-fei Wang[1ʘ], Yi Xu[1], Tian Xu[1], Pei-na Meng[1], Xiang-qi Wu[1], Zhi-ming Wu[1], Xiao-han Kong[1], Hai-bo Jia[1], De-lu Yin[3]\*, Lang Li[2]\*, Fei Ye[1]\***

**1** Department of Cardiology, Nanjing First Hospital Affiliated to Nanjing Medical University, Nanjing, 210006, China, **2** Department of Cardiology, The First Affiliated Hospital of Guangxi Medical University, Nanning, 530021, China, **3** Department of Cardiology, The First Hospital of Lianyungang Affiliated to Xuzhou Medical University, Haizhou District, Lianyungang, 222061, China

ʘ These authors contributed equally to this work.

\* doctor_ye@126.com (FY); drlilang1968@126.com (LL); druseyin@163.com (DY)

## Abstract

### Introduction

The natural outcome of coronary plaque in acute coronary syndrome (ACS) patients with chronic kidney disease (CKD) is unique, which can be analyzed quantitatively by optical flow ratio (OFR) software.

### Methods

A total of 184 ACS patients with at least one nonculprit subclinical atherosclerosis (NSA) detected by optical coherence tomography (OCT) at baseline and 1-year follow-up were divided into non-CKD group (n = 106, estimated glomerular filtration rate (eGFR) > 90 mL/(min×1.73 m²)) and mild CKD group (n = 78, 60 ≤ eGFR < 90 mL/(min×1.73 m²)). Changes of normalized total atheroma volume (TAVn) of NSA was the primary endpoint at the 1-year follow-up.

### Results

Patients with mild CKD showed more TAVn progression of NSA than non-CKD (p = 0.019) from baseline to the 1-year follow-up, which was mainly due to an increase in calcium TAVn (p<0.001). The morphological change in the maximal calcification thickness (p = 0.026) was higher and the change in the distance from the calcified surface to the contralateral coronary media membrane (ΔC-to-M) at the maximal cross-sectional calcium area was lower

**Data Availability Statement:** All relevant files in the manuscript are available from the figshare database (accession number(s) DOI: 10.6084/m9.figshare.25414681).

**Funding:** The study was supported by the JiangSu Provincial (China) Health Commission medical research project (ZDB2020029 to D.l Y) and the AstraZeneca Corp of China (ISSBRIL0361 to F. Y). The funders had no role in study design, data collection and analysis, decision to publish, or preparation of the manuscript.

**Competing interests:** none

(p<0.001) in mild CKD group than in non-CKD group. Mild CKD had more NSA related MACEs at the 5-year follow-up than non-CKD (30.8% vs. 5.8%, p = 0.045).

## Conclusions

Mild CKD patients had more plaque progression of NSA which showed the increase of calcium component with more protrusion into the lumen morphologically at the 1-year follow-up and a higher corresponding incidence of NSA-related MACEs at the 5-year follow-up.

## Trial registration

**Clinical Trial registration** ClinicalTrials.gov. NCT02140801. https://classic.clinicaltrials.gov/ct2/show/NCT02140801.

## Introduction

Cardiovascular disease is currently the leading cause of death in humans, an growing evidence shows that intensive lipid-lowering therapy (LLT) can stabilize and reverse coronary atherosclerotic plaque [1–5] and improve the clinical prognosis of patients [6,7]. However, for some special populations, such as those with diabetes and renal insufficiency, the therapeutic effect is not the same; even with similar statin-based LLT, the outcome of subclinical atherosclerotic lesions in these patients can differ [8–12]. In particular, in patients with chronic kidney disease (CKD), statin-based LLT appears to reduce cardiovascular mortality unsatisfactorily, possibly because the characteristics of atherosclerotic plaques in these patients are different from those in non-CKD patients [13–15].

The characteristics of coronary lesions in patients with advanced CKD or dialysis are characterized by calcification proliferation at the media level, unlike the coronary plaques in non-CKD patients, and with the aggravation of renal function, especially for patients with end-stage CKD or after dialysis, the coronary calcification reaches its peak severity [16–18]. However, there have been inconsistent conclusions about the nature of coronary plaque in patients with mild CKD; one study showed an increase in lipid content [18], while other studies showed that in such patients, medial calcification, vascular inflammation, and plaque neovascularization increased [19,20].

At present, intravascular imaging, such as intravascular ultrasound (IVUS) or optical coherence tomography (OCT), is usually used in studies of coronary plaques in vivo. In patients with CKD, only a few studies have used OCT for acute kidney injury, which may be caused by contrast agents [21], and most of them have focused on IVUS [21–23], even though it cannot accurately measure calcification lesions. Recently, artificial intelligence (AI) conception-based software for automatic measurement of coronary plaque components has been increasingly applied in the analysis of OCT data [10,24,25], which is named OFR for "optical flow ratio" (Pulse Medical Imaging Technology, Shanghai, Co., Ltd). The histological similarity of its plaque measurement has been demonstrated in clinical studies many times [24,26]. We applied this software to analyze nonculprit subclinical atherosclerosis (NSA) in patients with mild CKD or without CKD to explore the effects of mild renal insufficiency on natural outcomes of NSA at the 1-year follow-up and the clinical outcome of NSA-related major adverse cardiac events (MACEs) at the 5-year follow-up.

## Materials and methods

### Study population study design

All the data in this study came from one of our previous randomized controlled studies [27] (ClinicalTrials.gov. Number: NCT02140801), which started in 2014–05 and ended in 2018–03 of the recruitment periods for this study. Of the 352 patients with acute coronary syndrome whose culprit lesions underwent OCT-guided percutaneous coronary intervention (PCI) therapy, 184 also had at least one NSA (defined as OCT-measured plaque burden between 30% and 70%) without PCI treatment but underwent OCT detection at both baseline and the 1-year follow-up and met the inclusion criteria. If the patient had multiple NSAs, one of them was selected as the analysis target (the selection priority was in the following order: high-quality OCT image, location in the proximal or middle segment of left anterior descending branch, location in the proximal or middle segment of right coronary artery, and location in the proximal or middle segment of left circumflex) [25,28]. Clinical follow-up was performed for at least 5 years. All the enrolled patients were divided into mild CKD group and non-CKD groups based on baseline renal function, which was evaluated according to the recommendations of New Creatinine- and Cystatin C-Based Equations to Estimate GFR without Race (the CKD-EPI 2021 equation) in our retrospective study [29]. The estimated glomerular filtration rate (eGFR), which was calculated by adjusting serum creatinine (Scr) as eGFRcr (mL/(min×1.73 m$^2$)) = $142 \times (Scr/0.7)^{-0.241} \times 0.9938^{age} \times 1.012$ (if Scr≤0.7 mg per deciliter for female participants), or = $142 \times (Scr/0.7)^{-1.200} \times 0.9938^{age} \times 1.012$ (if Scr>0.7 mg per deciliter for female participants), or = $142 \times (Scr/0.9)^{-0.302} \times 0.9938^{age}$ (if Scr≤0.9 mg per deciliter for male participants), or = $142 \times (Scr/0.9)^{-1.200} \times 0.9938^{age}$ (if Scr>0.9 mg per deciliter for male participants) [30] was the major criterion of renal function assessment in our study. Staging renal function are performed as recommended in the guidelines of KDIGO criteria [31], which defines that G1 as increased (eGFR > 105 mL/(min×1.73 m$^2$)) or optimal (105 mL/(min×1.73 m$^2$) >eGFR >90 mL/(min×1.73 m$^2$)) renal function, G2 as mild CKD (89 mL/(min×1.73 m$^2$) >eGFR >60 mL/(min×1.73 m$^2$)), G3a as mild-moderate CKD (59 mL/(min×1.73 m$^2$) >eGFR >45 mL/(min×1.73 m$^2$)), G3b as moderate-severe CKD (44 mL/(min×1.73 m$^2$) >eGFR >30 mL/(min×1.73 m$^2$)), G4 as severe CKD (29 mL/(min×1.73 m$^2$) >eGFR >15 mL/(min×1.73 m$^2$)), and G5 as kidney failure (eGFR < 15 mL/(min×1.73 m$^2$)). Inclusion criteria: men and women were 18 years and older; native NSA was detected by OCT at baseline and at 1 year post PCI. Finally, a total of 106 patients were enrolled into non-CKD group with G1 renal function, and 78 patients were enrolled into mild-CKD group with G2 CKD. Exclusion criteria: incomplete OCT data or lack of 1-year OCT data at the same site, poor OCT image quality, and no NSA (shown in **Fig 1**). All patients signed informed consent before the PCI procedure, and the study was approved by the Ethics Committee of Nanjing First Hospital (Approval number: KY20131121-03). All the methods used in this study were in accordance with the relevant guidelines and regulations.

### Medical therapy and clinical follow-up

All patients received dual antiplatelet therapy (including aspirin 100 mg/d and adenosine diphosphate receptor blockers, clopidogrel 75 mg/d or ticagrelor 90 mg bid) for at least one year followed by single antiplatelet therapy with aspirin 100 mg/d for life. All patients were treated with statins (combined with/without ezetimibe depending on the low-density lipoprotein cholesterol (LDL-C) level) and had other risk factors (such as diabetes and hypertension) controlled at the discretion of the treating physician. The clinical follow-up was performed by the follow-up team of the Department of Cardiology of Nanjing First Hospital at 1, 3, 6 and 12

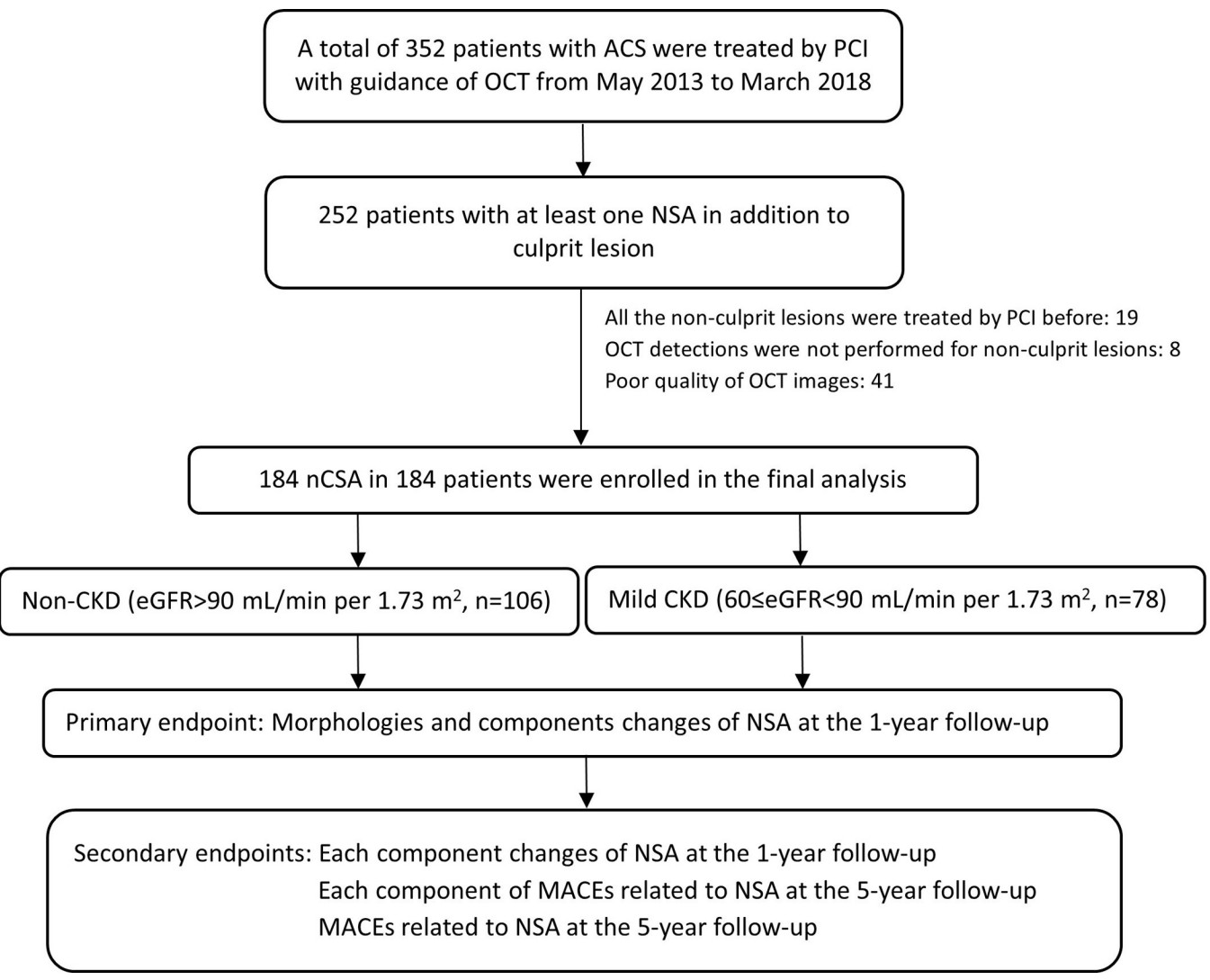

**Fig 1. Study flow chart.** Study flow chart for patient inclusion and grouping. ACS, acute coronary syndrome; CKD, chronic kidney disease; eGFR, estimated glomerular filtration rate; MACEs, major adverse cardiac events; NSA, nonculprit subclinical atherosclerosis; OCT, optical coherence tomography; PCI, percutaneous coronary intervention.

months and at 2, 3, 4 and 5 years after PCI in the outpatient department or by telephone and recorded in the case report form in detail.

## OCT image acquisition and analysis

OCT images were acquired by both ILUMIEN OPTIS and C7-XR (Lightlab Imaging Incorporated, Westford, MA) with a 2.7-F (Dragonfly OPTIS or Dragonfly Duo imaging catheter, Westford, MA) catheter automatic pullback system moving at a speed of 36 mm/s with continuous contrast injection for red blood cell removal after intracoronary nitroglycerine injection. All OCT images were stored in DICOM format and analyzed offline with dedicated software (OFR, Pulse Medical Imaging Technology, Shanghai, Co., Ltd.) [24], which measured the quantitative planimetry of plaque characteristics by automatic border detection followed by manual correction, and each area of lipid, fibrous, calcium and macrophage components was measured frame by frame to determine the corresponding volumes in the measured segment

[24]. Finally, the quantitative indicators of each plaque component were recorded in accordance with previous studies, as follows [3,10,25]: external elastic membrane area ($EEM_{area}$) was defined as the cross-sectional area of the EEM, lumen area ($Lumen_{area}$) was defined as the cross-sectional area of the lumen, normalized total atheroma volume (TAVn) was calculated as $\Sigma(EEM_{area}-Lumen_{area})$/number of frames of target segment×100, percent atheroma volume (PAV) was calculated as $\Sigma(EEM_{area}-Lumen_{area})/\Sigma EEM_{area}\times100$, lipid TAVn was defined as $\Sigma lipid_{area}$/number of frames of target segment×100), lipid PAV was defined as $\Sigma lipid_{area}/\Sigma EEM_{area}\times100$, fibrous TAVn was defined as $\Sigma fibrous_{area}$/number of frames of target segment×100, fibrous PAV was defined as $\Sigma fibrous_{area}/\Sigma EEM_{area}\times100$, calcium TAVn was defined as $\Sigma calcium_{area}$/number of frames of target segment×100, calcium PAV was defined as $\Sigma calcium_{area}/\Sigma EEM_{area}\times100$, macrophage TAVn was defined as $\Sigma macrophage_{area}$/number of frames of target segment×100, and macrophage PAV was defined as $\Sigma macrophage_{area}/\Sigma EEM_{area}\times100$. The change in TAVn (ΔTAVn) was defined as the TAVn at the 1-year follow-up minus the baseline TAVn, which could be derived from lipid ΔTAVn, fibrous ΔTAVn, calcium ΔTAVn or macrophage ΔTAVn. The change in PAV (ΔPAV) was defined as the PAV at the 1-year follow-up minus the baseline PAV, which could be derived from lipid ΔPAV, fibrous ΔPAV, calcium ΔPAV or macrophage ΔPAV. The thinnest fibrous cap thickness (TFCT) in the same segment was also measured, and the change in TFCT (ΔTFCT) was defined as the TFCT at the 1-year follow-up minus the baseline TFCT. OFR automatically measured the following in the analyzed segment: minimal luminal area (MLA), maximal calcification area (MaxCA), maximal calcification radian (MaxCR), defined as the maximal angulation of calcification, and maximal calcification thickness (MaxCT). The distance from the calcified surface to the contralateral coronary medium membrane (C-to-M) was measured at MCA. Before comparing any data, a consistency analysis of baseline and 1-year follow-up OCT images was required by manual multiaspect coregistration based on reproducible index side branches, known pullback speeds of OCT catheter during procedure and combined with angiographic images [10,25]. All the OFR and angiographic images were measured by two independent technicians who had no knowledge of the patient's information and obtained certification after training by Pulse Medical Imaging Technology, Shanghai, Co., Ltd.

## Consistency test of OCT and OFR measurements

Intraobserver and interobserver variability of the OCT image analysis by OFR were assessed for fifty randomly selected data points for plaque component qualitative analysis by the kappa statistic (for categorical variables: lipid, fibrous, calcium characteristics, and macrophage) or the intraclass correlation coefficient (ICC) (for continuous variables: TAVn, PAV, lipid TAVn, lipid PAV, fibrous TAVn, fibrous PAV, calcium TAVn, calcium PAV, macrophage TAVn, macrophage PAV, and TFCT) by the same technician at a 2-week interval or by 2 independent technicians. All kappa values were > 0.9 and all ICCs were > 0.9 in the consistency tests.

## Study endpoints

The primary observational endpoint was the difference in components and morphologies of NSA between ACS patients with normal and mild CKD at the 1-year follow-up. The secondary endpoints included the difference in each component of NSA at the 1-year follow-up, the MACEs related to NSA at the 5-year follow-up, and the difference in each component of MACEs (such as any cardiac death, myocardial infarction, or ischemia-driven revascularization) related to NSA at the 5-year follow-up between the two groups.

## Statistical analysis

All continuous variables are described as the mean ± standard deviation for normally distributed data or median (interquartile range) for nonnormally distributed data, which were compared between groups using Student's t test or a nonparametric test, respectively. Categorized variables are described in terms of percentages and were analyzed using the chi-square test. MACE-free survival curves of patients with or without mild CKD were drawn using the Kaplan–Meier method and were compared by the log-rank test. All statistical tests were two-tailed, and the significance level was set at 0.05. All analyses were performed using SPSS version 26.0 software (IBM, Armonk, New York), and box diagrams were drawn using R software for Windows version 4.1.2 (https://www.r-project.org/).

## Results

### Patients' clinical characteristics at baseline and 1-year follow-up

All the patients' characteristics, including each risk factor for coronary artery disease, baseline medical therapy, and each corresponding serum marker, such as lipid profiles and renal function index such as Scr and eGFR at baseline and 1-year follow-up, are summarized in **Table 1**. The baseline data, including the prevalence of hypertension, diabetes mellitus, dyslipidemia, current smoking, lipid profile and medical therapy, were comparable between the two groups, except that the average age (69.65±8.15 vs. 62.42±10.80, $p<0.001$) and Scr level (86.00 (78.85, 98.25) μmol/L vs. 68.00 (58.00, 75.60) μmol/L, $p<0.001$) were higher but eGFR (76.99 (67.31, 84.25) mL/(min×1.73 m$^2$) vs. 105.67 (96.00, 116.29) mL/(min×1.73 m$^2$), $p<0.001$) was significantly lower in mild CKD than in non-CKD. Compared with baseline, each component of the lipid profile, such as total cholesterol, LDL-C, high-density lipoprotein cholesterol (HDL-C), non-HDL-C, and triglycerides, was decreased significantly in both groups, but no difference was found between the two groups at the 1-year follow-up. Scr (87.00 (73.50, 98.40) μmmol/L vs. 86.00 (78.85, 98.25) μmmol/L, $p>0.05$) and eGFR (75.24 (65.10, 83.53) mL/(min×1.73 m$^2$ vs. 76.99 (67.31, 84.25) mL/(min×1.73 m$^2$, $p>0.05$) did not change significantly at the 1-year follow-up compared with baseline in mild CKD. A small deterioration in status, manifested by an increase in Scr (68.00 (58.00, 75.60) μmmol/L vs. 69.20 (58.88, 78.32) μmmol/L, $p<0.05$) and a decrease in eGFR (105.67 (96.00, 116.29) mL/(min×1.73 m$^2$) vs. 98.23 (90.05, 113.73) mL/(min×1.73 m$^2$), $p<0.05$), was seen at 1 year in non-CKD group compared with baseline. Even so, Scr (87.00 (73.50, 98.40) μmmol/L vs. 69.20 (58.88, 78.32) μmmol/L, $p<0.001$) was still higher and eGFR (75.24 (65.10, 83.53) mL/(min×1.73 m$^2$) vs. 98.23 (90.05, 113.73) mL/(min×1.73 m$^2$), $p<0.001$) was still lower in mild CKD than in non-CKD at the 1-year follow-up compared with baseline.

### OCT data analysis by OFR measurements

Only one NSA lesion was analyzed for each patient in our study. All the components of NSA measured by OFR at baseline and at the 1-year follow-up are summarized in **Table 2**. Baseline data of NSA, such as TAVn, PAV, and each component of NSA, were comparable between the two groups. Both TAVn (120.37±44.12 mm$^3$ vs. 125.16±45.30 mm$^3$, $p<0.05$) and PAV (43.39 ±8.59% vs., 45.15±7.76%, $p<0.05$) increased significantly from baseline to 1-year follow-up in mild CKD. In contrast, there was no significant change in TAVn or PAV from baseline to 1-year follow-up in non-CKD, which finally caused TAVn to be significantly higher in mild CKD than in non-CKD (125.16±45.30 mm$^3$ vs. 110.45±39.12 mm$^3$, $p = 0.019$) at the 1-year follow-up. Lipid components such as lipid TAVn and lipid PAV nonsignificantly decreased at 1 year compared with baseline, although fibrous components such as fibrous TAVn (79.69

**Table 1. Patients' clinical characteristics at baseline and 1-year follow-up.**

| | Total (n = 184) | Non-CKD (n = 106) | Mild CKD (n = 78) | P Value |
|---|---|---|---|---|
| Age, yrs | 65.45±10.39 | 62.42±10.80 | 69.65±8.15 | <0.001 |
| Men, n(%) | 139(75.5) | 84(79.2) | 55(70.5) | 0.173 |
| BMI | 24.94±2.86 | 24.95±2.85 | 24.94±2.88 | 0.994 |
| Hypertension, n(%) | 126(68.5) | 68(64.2) | 58(74.4) | 0.141 |
| Diabetes mellitus, n(%) | 49(26.6) | 24(22.6) | 25(32.1) | 0.154 |
| Dyslipidemia, n(%) | 122(66.3) | 72(67.9) | 50(64.1) | 0.588 |
| Smoking, n(%) | 59(32.1) | 39(36.8) | 20(25.6) | 0.109 |
| Previous PCI, n(%) | 27(14.7) | 13(12.3) | 14(17.9) | 0.282 |
| Baseline medical therapy | | | | |
| Statin, n(%) | 184(100.0) | 106(100.0) | 78(100.0) | NA |
| Ezetimibe, n(%) | 67(36.4) | 38(35.8) | 29(37.2) | 0.853 |
| Antiplatelet therapy, n(%) | 184(100.0) | 106(100.0) | 78(100.0) | NA |
| ACEI/ARB, n(%) | 108(58.7) | 58(54.7) | 50(64.1) | 0.201 |
| CCB, n(%) | 49(26.7) | 28(26.4) | 21(26.9) | 0.939 |
| Beta-blocker, n(%) | 98(53.3) | 55(51.9) | 43(55.1) | 0.663 |
| Baseline lipid profile | | | | |
| Total cholesterol, mmol/L | 3.69(3.11,4.54) | 3.65(3.06,4.60) | 3.74(3.14,4.44) | 0.533 |
| LDL-C, mmol/L | 2.05(1.65,2.80) | 2.01(1.62,2.97) | 2.07(1.70,2.63) | 0.837 |
| HDL-C, mmol/L | 0.99(0.87,1.18) | 0.98(0.86,1.18) | 1.01(0.87,1.20) | 0.415 |
| non-HDL-C, mmol/L | 2.74(2.13,3.40) | 2.70(2.15,3.44) | 2.76(2.11,3.31) | 0.890 |
| Triglycerides, mmol/L | 1.40(1.00,1.95) | 1.40(1.00,2.03) | 1.39(1.01,1.95) | 0.831 |
| Scr, μmmol/L | 75.00(65.00,83.75) | 68.00(58.00,75.60) | 86.00(78.85,98.25) | <0.001 |
| eGFR, mL/(min$^*$1.73m$^2$) | 92.75(80.84,108.31) | 105.67(96.00,116.29) | 76.99(67.31,84.25) | <0.001 |
| Serum calcium, mmol/L | 2.23±0.12 | 2.21±0.11 | 2.24±0.14 | 0.783 |
| Serum phosphorus, mmol/L | 1.16±0.22 | 1.14±0.21 | 1.18±0.22 | 0.345 |
| Serum uric acid, μmmol/L | 338.64±91.51 | 321.54±85.02 | 362.26±95.40 | 0.003 |
| 1-year Follow up lipid profile | | | | |
| Total cholesterol, mmol/L | 3.37(3.01,3.88) [***] | 3.34(3.00,3.84) [**] | 3.43(3.01,3.96) [**] | 0.550 |
| LDL-C, mmol/L | 1.70(1.43,2.13) [***] | 1.69(1.42,2.10) [***] | 1.75(1.43,2.22) [***] | 0.504 |
| HDL-C, mmol/L | 1.16(1.01,1.34) [***] | 1.14(1.02,1.34) [***] | 1.17(0.98,1.34) [***] | 0.850 |
| non-HDL-C, mmol/L | 2.19(1.84,2.66) [***] | 2.18(1.77,2.54) [**] | 2.26(1.88,2.75) [***] | 0.511 |
| Triglycerides, mmol/L | 1.15(0.93,1.65) [***] | 1.18(0.94,1.68) [**] | 1.14(0.93,1.59) [***] | 0.591 |
| Scr, μmmol/L | 76.00(63.50,87.00) | 69.20(58.88,78.32) [*] | 87.00(73.50,98.40) | <0.001 |
| eGFR, mL/(min$^*$1.73m$^2$) | 91.69(76.04,105.12) | 98.23(90.05,113.73) [*] | 75.24(65.10,83.53) | <0.001 |
| Change in lipid profile between index and 1-year follow-up | | | | |
| ΔTotal cholesterol, mmol/L | -0.24(-1.01, 0.45) | -0.26(-1.02, 0.51) | -0.23(-1.01, 0.26) | 0.467 |
| ΔLDL-C, mmol/L | -0.35(-0.88, 0.17) | -0.35(-0.89, 0.17) | -0.36(-0.91, 0.16) | 0.833 |
| ΔLDL-C,% | -17.89(-35.42,9.66) | -18.60(-35.97,10.95) | -17.31(-35.29,6.52) | 0.998 |
| LDL-C<1.4mmol/L, % | 41(22.3) | 23(21.7) | 18(23.1) | 0.841 |
| Δnon-HDL-C, mmol/L | -0.48(-1.10,0.25) | -0.48(-1.15,0.30) | -0.42(-1.05,0.16) | 0.678 |
| ΔHDL-C, mmol/L | 0.15 (0.02, 0.28) | 0.17(0.06, 0.31) | 0.10(0.01, 0.23) | 0.072 |
| ΔTriglycerides, mmol/L | -0.20(-0.63, 0.17) | -0.19(-0.59, 0.22) | -0.24(-0.73, 0.12) | 0.287 |
| ΔScr, μmmol/L | 2.10(-0.60,8.00) | 3.00(-4.63,8.08) | 1.00(-9.00,4.95) | 0.059 |
| ΔeGFR, mL/(min$^*$1.73m$^2$) | -3.67(-11.18,8.25) | -5.91(-16.46,6.05) | -1.21(-7.83,9.73) | 0.002 |

[***] P<0.001

[**] P<0.01

[*] P<0.05 (these values were compared between baseline and 1-year follow-up.)

Values are expressed as the median (interquartile range) for continuous variables with abnormal distribution and described as the mean ± standard deviation with normal distribution, or frequency (percentage) for categorical variables in the table. ACEI, angiotensin-converting enzyme inhibitor; ARB, angiotensin receptor blocker; BMI, body mass index; CCB, calcium channel blocker; eGFR, estimated glomerular filtration rate; HDL-C, high-density lipoprotein cholesterol; LDL-C, low-density lipoprotein cholesterol; PCI, percutaneous coronary intervention; Scr, serum creatinine.

**Table 2. Components of NSA changes at the 1-year follow-up measured by OFR.**

| | Non-CKD (n = 106) | Mild CKD (n = 78) | P Value |
|---|---|---|---|
| TAVn, mm$^3$ | | | |
| baseline | 109.52±37.10 | 120.37±44.12 | 0.072 |
| 1-year follow-up | 110.45±39.12 | 125.16±45.30[*] | 0.019 |
| ΔTAVn | 0.79[-8.19,8.54] | 5.25[-3.85,12.10] | 0.067 |
| PAV, % | | | |
| baseline | 43.88±8.91 | 43.39±8.59 | 0.708 |
| 1-year follow-up | 43.85±8.45 | 45.15±7.76[*] | 0.287 |
| ΔPAV | 0.00[-2.45,2.45] | 0.85[-1.73,4.00] | 0.063 |
| Lipid TAVn, mm$^3$ | | | |
| baseline | 16.72[8.12,26.71] | 20.08[9.07,35.41] | 0.109 |
| 1-year follow-up | 15.60[7.00,27.37] | 19.06[7.93,35.06] | 0.151 |
| Lipid ΔTAVn | -0.55[-7.30,3.53] | -0.10[-7.24,6.98] | 0.536 |
| Lipid PAV, % | | | |
| baseline | 16.60[9.00,25.20] | 18.65[10.00,26.70] | 0.359 |
| 1-year follow-up | 15.00[8,50,23.25] | 16.95[8.25,25.33] | 0.409 |
| Lipid ΔPAV | -0.94[-6.60,2.63] | -0.55[-6.25,3.20] | 0.734 |
| Fibrous TAVn, mm$^3$ | | | |
| baseline | 66.94[54.35,90.44] | 75.51[55.81,91.43] | 0.241 |
| 1-year follow-up | 69.52[55.67,94.76][*] | 79.69[60.65,96.63][***] | 0.203 |
| Fibrous ΔTAVn | 2.60[-4.10,7.64] | 4.41[-2.30,9.86] | 0.173 |
| Fibrous PAV, % | | | |
| baseline | 69.50[59.30,75.70] | 65.95[55.30,74.40] | 0.153 |
| 1-year follow-up | 71.10[63.63,78.63][**] | 67.90[55.55,76.80][*] | 0.102 |
| Fibrous ΔPAV | 2.00[-2.35,7.53] | 1.20[-3.20,6.90] | 0.369 |
| Calcium TAVn, mm$^3$ | | | |
| baseline | 0.47[0.02,2.38] | 0.53[0.04,3.19] | 0.402 |
| 1-year follow-up | 0.69[0.04,2.60][**] | 1.39[0.13,6.35][***] | 0.021 |
| Calcium ΔTAVn | 0.05[-0.12,0.76] | 0.56[0.05,3.03] | <0.001 |
| Calcium PAV, % | | | |
| baseline | 0.35[0.00,2.13] | 0.40[0.10,2.53] | 0.514 |
| 1-year follow-up | 0.55[0.00,2.28][**] | 0.70[0.10,5.03][***] | 0.165 |
| Calcium ΔPAV | 0.10[-0.10,0.70] | 0.28[0.00,2.03] | 0.015 |
| Macrophage TAVn, mm$^3$ | | | |
| baseline | 0.24[0.08,0.66] | 0.41[0.10,0.94] | 0.229 |
| 1-year follow-up | 0.15[0.06,0.60][*] | 0.33[0.12,0.73] | 0.045 |
| Macrophage ΔTAVn | -0.04[-0.27,0.10] | 0.00[-0.23,0.20] | 0.276 |
| Macrophage PAV, % | | | |
| baseline | 0.30[0.10,0.70] | 0.30[0.10,0.70] | 0.565 |
| 1-year follow-up | 0.20[0.10,0.60][*] | 0.30[0.10,0.73] | 0.044 |
| Macrophage ΔPAV | -0.10[-0.30,0.10] | 0.00[-0.20,0.20] | 0.188 |
| TFCT, μm | | | |
| baseline | 114.00[75.00,159.50] | 106.00[77.00,134.00] | 0.524 |
| 1-year follow-up | 137.00[84.75,179.25][**] | 123.00[79.00,170.25][*] | 0.394 |
| ΔTFCT | 19.00[-16.25,70.00] | 17.00[-25.50,72.00] | 0.939 |
| MLA, mm$^2$ | | | |
| baseline | 4.37[3.05,6.19] | 4.58[3.02,7.64] | 0.417 |
| 1-year follow-up | 4.21[2.90,5.75][*] | 4.37[2.85,6.64][*] | 0.548 |
| ΔMLA | -0.17[-0.92,0.40] | -0.18[-1.34,0.40] | 0.602 |

[***]$P<0.001$

[**]$P<0.01$

[*]$P<0.05$ (these values were compared between baseline and 1-year follow-up).

Values are expressed as the median (interquartile range) for continuous variables with abnormal distribution and described as the mean ± standard deviation with normal distribution, or frequency (percentage) for categorical variables in the table. MLA, minimal luminal area; NSA, non-culprit subclinical atherosclerosis; OFR, optical flow ratio; PAV, percent atheroma volume; TAVn, normalized total atheroma volume; TFCT, thinnest fibrous cap thickness.

[60.65, 96.63] mm$^3$ vs. 75.51 [55.81, 91.43] mm$^3$, p<0.001), fibrous PAV (67.90 [55.55, 76.80]% vs. 65.95 [55.30, 74.40]%, p<0.05) in mild CKD and fibrous TAVn (69.52 [55.67, 94.76] mm$^3$ vs. 66.94 [54.35, 90.44] mm$^3$, p<0.05) and fibrous PAV (71.10 [63.63, 78.63]% vs. 69.50 [59.30, 75.70]%, p<0.01) in non-CKD increased significantly at 1 year compared with baseline, but no significant difference in final fibrous TAVn, PAV, fibrous ΔTAVn, or fibrous ΔPAV was found between two groups at 1 year. Calcium data such as calcium TAVn (1.39 [0.13, 6.35] mm$^3$ vs. 0.53 [0.04, 3.19] mm$^3$, p<0.001), calcium PAV (0.70 [0.10, 5.03]% vs. 0.40 [0.10, 2.53]%, p<0.001) in mild CKD and calcium TAVn (0.69[0.04, 2.60] mm$^3$ vs. 0.47[0.02, 2.38] mm$^3$, p<0.01), calcium PAV (0.55[0.00, 2.28]% vs. 0.35[0.00, 2.13]%, p<0.01) in non-CKD were increased significantly at 1 year compared with baseline, which were caused by the significant increases in calcium ΔTAVn (0.56 [0.05, 3.03] mm$^3$ vs. 0.05 [-0.12, 0.76] mm$^3$, p<0.001) and calcium ΔPAV (0.28 [0.00, 2.03]% vs. 0.10 [-0.10, 0.70]%, p = 0.015). Although macrophage-related data showed a declining trend in both groups, only macrophage TAVn (0.15 [0.06, 0.60] mm$^3$ vs. 0.24 [0.08, 0.66] mm$^3$, p<0.05) and macrophage PAV (0.20 [0.10, 0.60]% vs. 0.30 [0.10, 0.70]%, p<0.05) showed a significant decrease at 1 year compared with baseline, which caused macrophage TAVn (0.15 [0.06, 0.60] mm$^3$ vs. 0.33 [0.12, 0.73] mm$^3$, p = 0.045) and macrophage PAV (0.20 [0.10, 0.60]% vs. 0.30 [0.10, 0.73]%, p = 0.44) to be lower at 1 year in non-CKD than mild CKD. TFCT increased in both non-CKD (137.00 [84.75, 179.25] μm vs. 114.00 [75.00, 159.50] μm, p<0.01) and mild CKD (123.00 [79.00, 170.25] μm vs. 106.00 [77.00, 134.00] μm, p<0.05), but ΔTFCT and final TFCT at 1 year were not significantly different between the two groups. Although MLA in both groups decreased significantly (p<0.05) after one year, there was still no significant difference between the two groups (shown in **Table 2** and **Fig 2**).

## Calcification morphology analysis by OFR measurements

Because the most significant change in both groups was calcification rather than the other two main components, lipid and fibrous, in the NSA at the 1-year follow-up, we carefully analyzed the characteristics of the calcification morphology, including MaxCA, MaxCR, MaxCT and C-to-M, in the target vascular section (shown in **Fig 3** and **Table 3**). MaxCA significantly increased in mild CKD (0.83 [0.20, 1.95] mm$^2$ vs. 0.56 [0.09, 1.45] mm$^2$, p<0.001), but not in non-CKD (0.51 [0.09, 1.28] mm$^2$ vs. 0.44[0.04, 1.06] mm$^2$, p>0.05), which was manifested in the degrees of MaxCR (49.30 [25.60, 90.00]˚ vs. 43.20 [18.10, 73.90]˚, p<0.001) and the thickness of MaxCT (0.74 [0.42, 1.08] mm vs. 0.59 [0.25, 1.01] mm, p<0.01) being significantly increased in mild CKD but not non-CKD at 1 year compared with baseline. Further analysis of calcification morphology showed that the increase in ΔMaxCT (0.09 [-0.04, 0.32] mm vs. 0.00 [-0.12, 0.16] mm, p = 0.026) was more significant in mild CKD than in non-CKD, but there was no significant difference in ΔMaxCR between the two groups. C-to-M decreased significantly in mild CKD (3.30±0.74 mm vs. 3.45±0.83 mm, p<0.01) and no significant changes in non-CKD (3.16±0.66 mm vs. 3.15±0.69 mm, p>0.05) at 1 year compared with baseline, and finally ΔC-to-M decreased more in mild CKD than in non-CKD (-0.12 [-0.31, 0.00] mm vs. 0.00 [-0.10, 0.08] mm, p<0.001).

## Exploratory analysis of NSA-related MACEs between the two groups

The 5-year clinical follow-up showed that there were more NSA-related MACEs in mild CKD than non-CKD (30.8% vs. 5.8%, p = 0.045) by Kaplan–Meier analysis (shown in **Fig 4**). As shown in the figure, the two curves intersected near the 3-year follow-up. Landmark analysis showed that there was no significant difference in the NSA-related MACEs between the two groups within nearly 3 years after PCI, while the MACEs in mild CKD became significantly

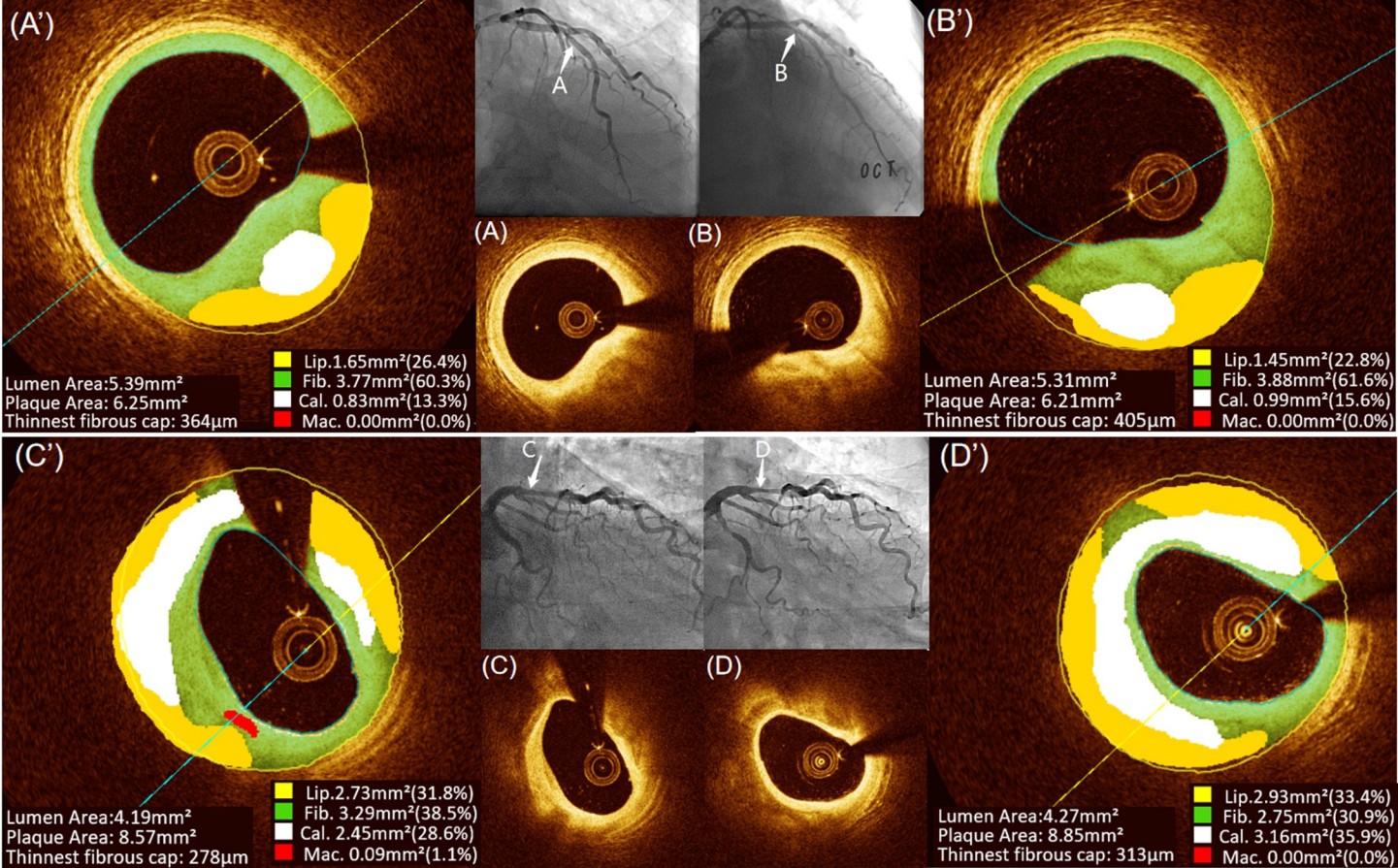

**Fig 2. Typical case of natural changes of coronary plaque components in patients with or without mild CKD.** Representative cross-sectional OCT image in a patient with non-CKD demonstrating similar lumen area, calcium and mild reduction in lipid component from baseline (A: observational point in angiography, (A): original OCT image, (A′): OFR measurement for each component in plaque) to 1-year follow-up (B: same observational point in angiography; (B): same point of OCT image at the 1-year follow-up; (B′): OFR measurement at the same point at the 1-year follow-up). Representative cross-sectional OCT image in another patient with mild CKD demonstrating mild reduction in lumen area and increase in calcium component from baseline (C: observational point in angiography, (C): original OCT image, (C′): OFR measurement for each component in plaque) to 1-year follow-up (D: same observational point in angiography; (D): same point of OCT image at the 1-year follow-up; (D′): OFR measurement at the same point at the 1-year follow-up). CKD, chronic kidney disease; OCT, optical coherence tomography; OFR, optical flow ratio; PAV, percent atheroma volume.

more common than those in non-CKD (25.7% vs. 0.7%, p = 0.006) during 3 to 5 years (shown in **Fig 5**). Further analysis of the components of MACEs related to NSA revealed that the whole difference was due to ischemia-driven revascularization (24.6% vs. 5.8%, p = 0.365), cardiac death (7.4% vs. 0%, p = 0.023) and myocardial infarction (3.8% vs. 1.1%, p = 0.40) in mild CKD compared with non-CKD.

## Discussion

The main findings of this retrospective observational study are as follows: 1) Even in ACS patients treated by similar lipid-lowering statin therapy, the TAVn of NSA increased significantly, with the calcium component increasing in mild CKD compared with that in non-CKD at the 1-year follow-up; 2) the morphological changes in calcium lesions showed greater MaxCT and more lumen protrusion in mild CKD compared to non-CKD; and 3) MACEs related to NSA were significantly higher in mild CKD than in non-CKD at the 5-year follow-up even with the same LLT at 1 year.

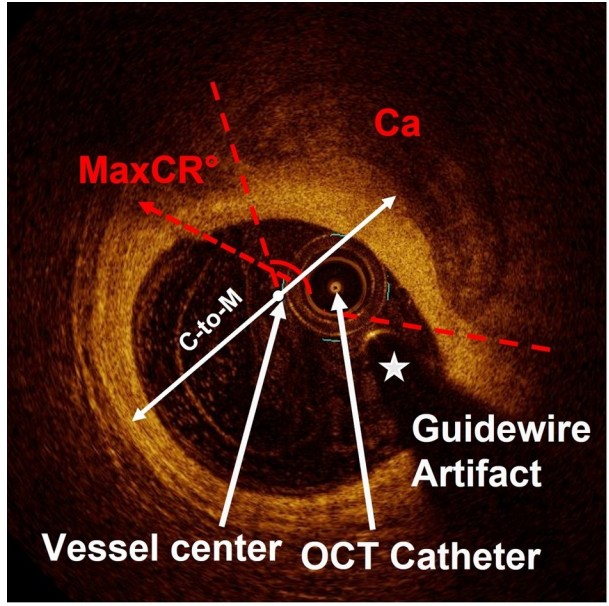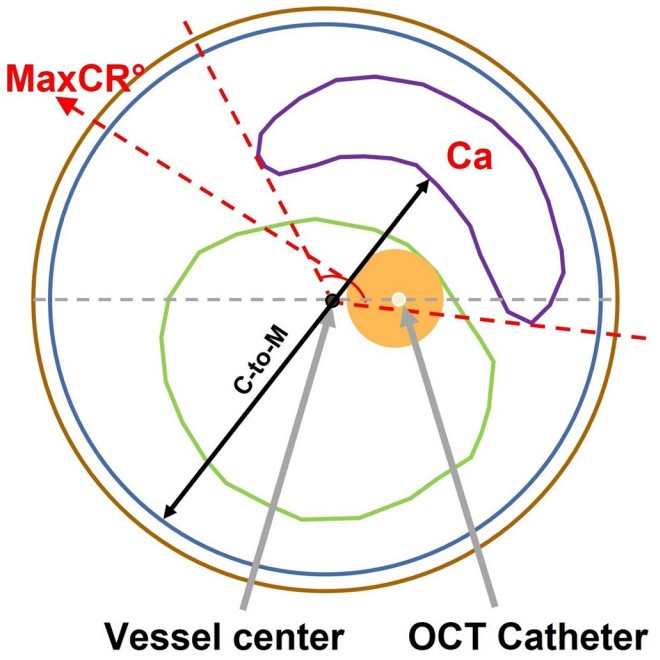

**Fig 3. Graphic measurement.** Graphic representation for data measurement by OFR software. C-to-M, the distance from the calcified surface to the contralateral coronary media membrane; MaxCR, maximal calcification radian; MaxCT, maximal calcification thickness; OCT, optical coherence tomography.

**Table 3. Morphological changes of calcification of NSA at the 1-year follow-up.**

|  | Non-CKD (n = 106) | Mild CKD (n = 78) | P Value |
|---|---|---|---|
| MaxCA, mm$^2$ |  |  |  |
| baseline | 0.44[0.04,1.06] | 0.56[0.09,1.45] | 0.443 |
| 1-year follow-up | 0.51[0.09,1.28] | 0.83[0.20,1.95] [***] | 0.069 |
| ΔMaxCA | 0.00[-0.16,0.26] | 0.16[-0.04,0.47] | 0.010 |
| MaxCR,˚ |  |  |  |
| baseline | 42.20[11.60,66.90] | 43.20[18.10,73.90] | 0.325 |
| 1-year follow-up | 44.20[14.10,72.90] | 49.30[25.60,90.00] [***] | 0.082 |
| ΔMaxCR | 0.00[-10.00,18.10] | 9.05[-8.50,26.10] | 0.230 |
| MaxCT, mm |  |  |  |
| baseline | 0.60[0.17,0.97] | 0.59[0.25,1.01] | 0.634 |
| 1-year follow-up | 0.66[0.29,0.98] | 0.74[0.42,1.08] [**] | 0.161 |
| ΔMaxCT | 0.00[-0.12,0.16] | 0.09[-0.04,0.32] | 0.026 |
| C-to-M, mm |  |  |  |
| baseline | 3.15±0.69 | 3.45±0.83 | 0.101 |
| 1-year follow-up | 3.16±0.66 | 3.30±0.74 [**] | 0.255 |
| ΔC-to-M | 0.00[-0.10,0.08] | -0.12[-0.31,0.00] | <0.001 |

[***] $P<0.001$

[**] $P<0.01$

[*] $P<0.05$ (these values were compared between baseline and 1-year follow-up).

Values are expressed as the median (interquartile range) for continuous variables with abnormal distribution and described as the mean ± standard deviation with normal distribution, or frequency (percentage) for categorical variables in the table. C-to-M: distance from calcified surface to contralateral coronary medium membrane at MaxCA; MaxCA: maximal calcification area; MaxCR: maximal calcification radian; MaxCT: maximal calcification thickness; NSA, non-culprit subclinical atherosclerosis.

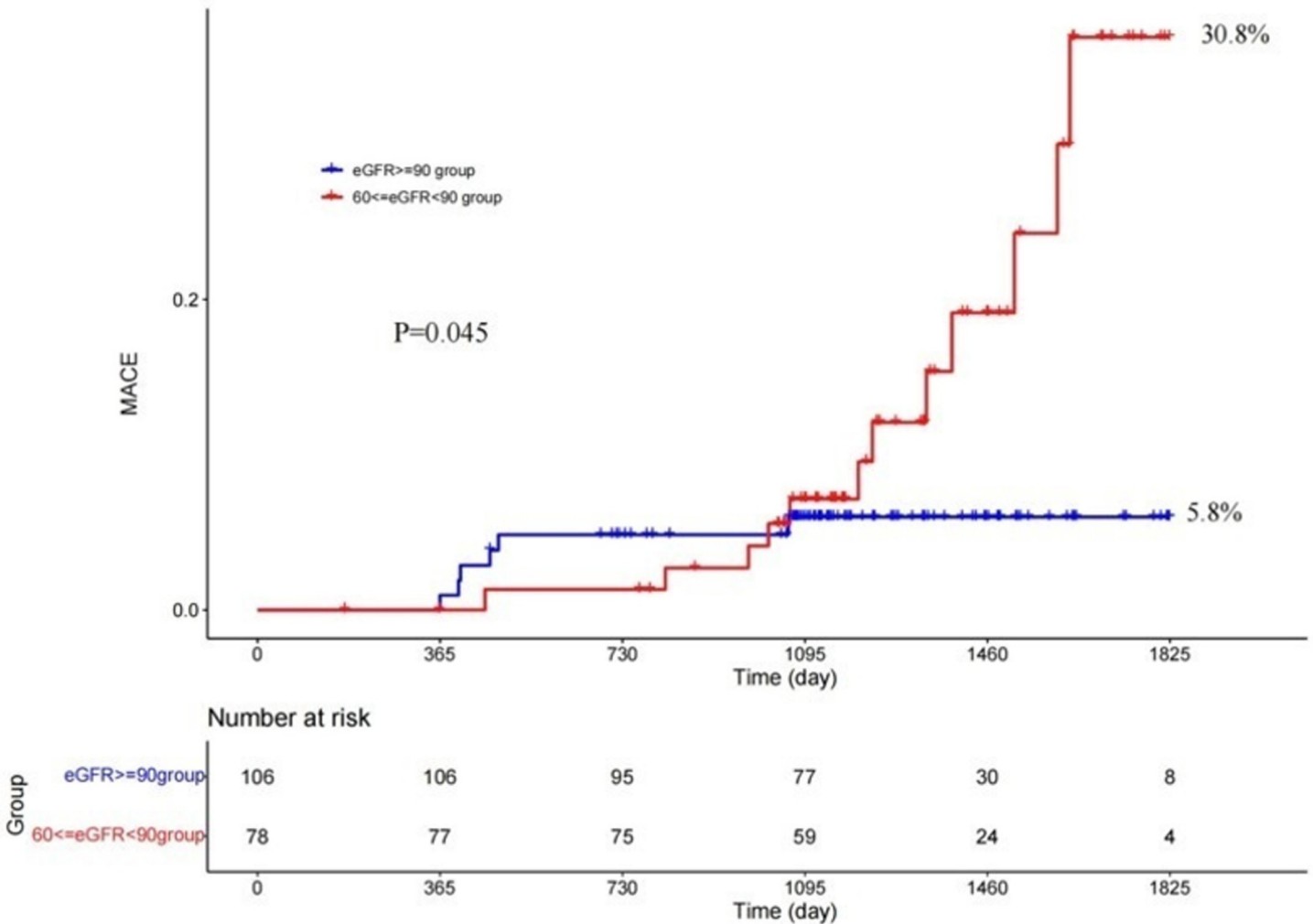

**Fig 4. MACE related to NSA at the 5-year follow-up.** eGFR, estimated glomerular filtration rate; MACEs, major adverse cardiac events.

An early study showed that eGFR was negatively correlated with the lipid volume of culprit lesions on IVUS detection but was positively associated with fibrous volume in nonhemodialysis CKD patients with stable angina [32]. Discrepant findings were found in another study that focused on nonculprit lesions with nonhemodialysis CKD and found that CKD patients with eGFR<60 mL/(min×1.73 m$^2$) had a higher lipid index with a higher prevalence of calcium [33]. Many studies on vascular calcification in patients with CKD focused on patients with advanced renal insufficiency or end-stage CKD with or without undergoing dialysis [34–40], most of which were cross-sectional studies comparing them with patients without CKD [41,42]. Several studies have suggested that the progression of coronary artery calcification in patients with moderate to severe CKD with or without continuous dialysis is significantly accelerated and is closely related to the subsequent increase in cardiac events [34,41]. However, the natural outcome of NSA in patients with mild renal insufficiency (60≤eGFR<90 mL/(min×1.73 m$^2$)) is not fully understood. Our study showed that even with a similar LLT, the TAVn of the NSA, which was detected by OCT and measured using novel OFR software that can measure plaque components automatically with a modern AI algorithm [10,24,25], increased significantly more in mild CKD than in non-CKD from baseline to the 1-year

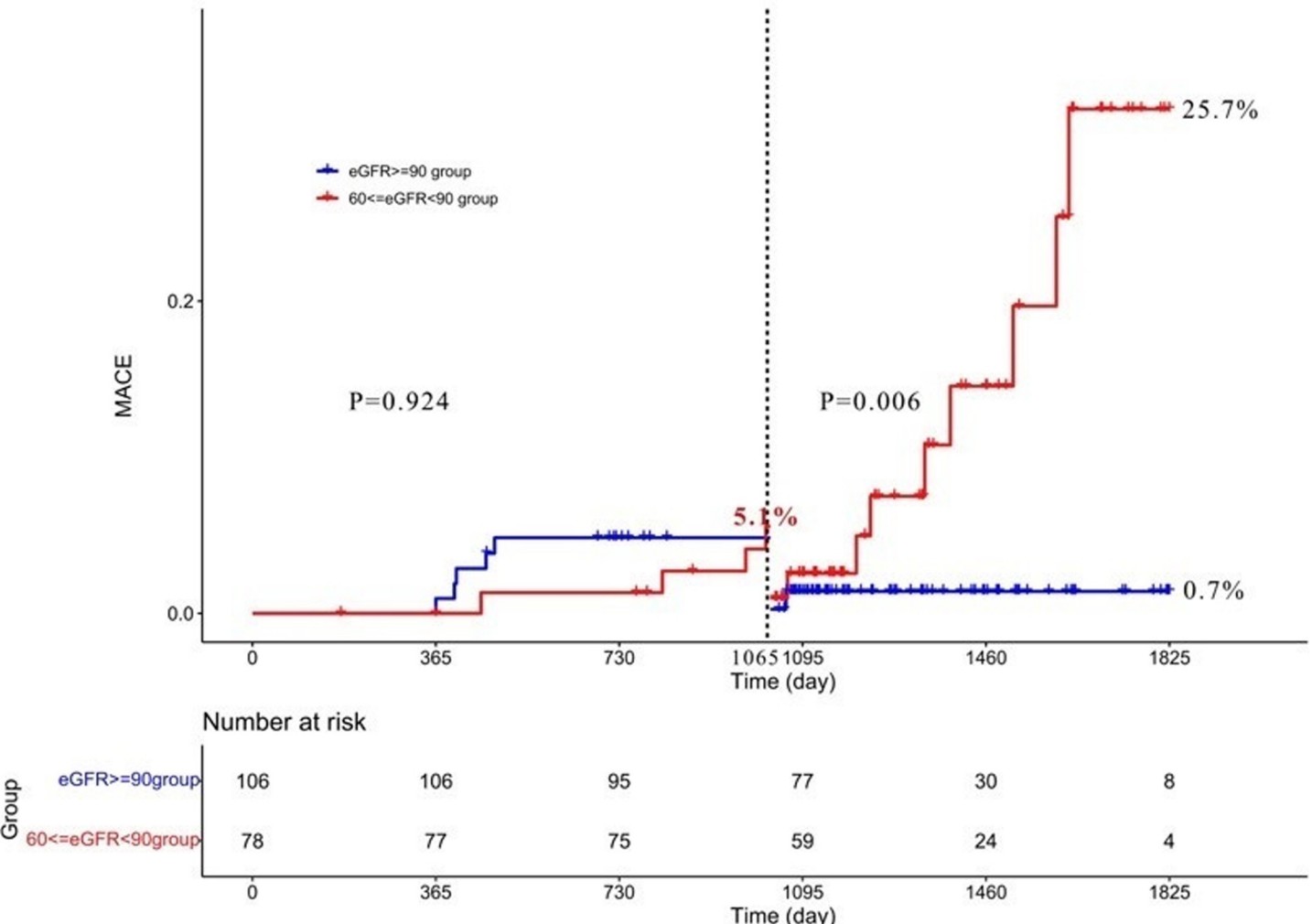

**Fig 5. Landmarker analysis of MACE related to NSA at the 5-year follow-up.** Landmark analysis showed that there was no significant difference in the NSA-related MACEs between the two groups within nearly 3 years after PCI, while the MACEs in mild CKD became significantly more common than those in non-CKD. eGFR, estimated glomerular filtration rate; MACEs, major adverse cardiac events.

follow-up. Quantitative analysis of changes in NSA components showed that fibrous and calcium components increased from baseline in both groups, but the degree of increase in calcium components was more significant in mild CKD than in non-CKD, while the degree of increase in fibrous components was similar in both groups. The lipid component decreased in both groups, with similar reductions in LDL-C levels, at the 1-year follow-up. Another interesting finding was that inflammatory components, such as macrophage TAVn and PAV, decreased significantly in mild CKD compared to non-CKD, which showed no significant changes from baseline to 1 year. This finding is consistent with the theory from previous studies that local inflammation induces coronary calcification; in other words, the reduction in inflammatory markers in plaques can delay the progression of calcification [20,39].

Although many factors are associated with the progression of coronary artery calcification, such as age, smoking, diabetes, and CKD [43,44], some studies have demonstrated that media coronary calcification was correlated with severe CKD or kidney failure (G4-5) patients [40,41,45], however, other studies in mild CKD patients or patients with normal renal function

(G1-2) have shown that atherosclerosis was dominant and more manifested by intimal calcification [46,47], in which most of these studies used computerized tomography images for analysis [41,42,44], and a few of them used intracoronary imaging such as OCT [43,48,49]. Our study found that patients with mild CKD may have calcification growth in both intima and media of coronary arteries, which may be related to the involvement of multiple factors [35]. Currently, in vivo studies, even with the use of more advanced intravascular imaging technology, there is still no clear quantitative method to analyze intimal and media calcification [48,49], and it is more difficult to study the natural outcome of plaque morphology. In our study, we used a novel method to analyze the changes in calcified morphology by OFR software, which measured ΔMaxCT (indicating the thickness changes of calcification) and ΔC-to-M (indicating the direction of calcification changes). Even at the 1-year follow-up, we found that MaxCA was significantly higher in mild CKD than in non-CKD, which was mainly due to the thickness of calcification increasing significantly in mild CKD compared to that in non-CKD, but no significant difference in the calcification angulation were found between the two groups at baseline and 1 year, even though mild CKD had a significant increase in MaxCR from baseline to 1 year. Morphological analysis showed that more protrusion into the lumen of calcification was found in mild CKD than in non-CKD, though the calcification growth may have been more toward the lumen, which indicated that intimal calcification increased more in mild CKD indirectly.

When the eGFR is lower than 60–75 mL/(min×1.73 m$^2$), the probability of developing coronary artery disease increases linearly [50,51]. The incidence of cardiac events at the 5- to 10-year follow-up in one study was significantly higher in CKD patients than non-CKD patients, even after adjusting for typical common risk factors for coronary artery disease (such as diabetes and hypertension) [52]. Standard clinical guidelines have considered CKD to be a "regulatory factor", which is independent of the traditional risk factors for coronary artery disease [53]. Previous clinical studies on the reduction of cardiovascular events have shown that LLT is only effective in patients with early CKD [13,54] and is less effective in patients with advanced CKD (especially those already on dialysis). One of the major findings in our study is that MACE related to NSA at the 5-year follow-up was significantly higher in the CKD group than in the non-CKD group, even in patients with mild CKD who were treated with similar LLT (based on statins) and had a similar reduction in LDL-C at 1 year. Another interesting finding is that there was no significant difference in NSA-related MACE between the two groups within 3 years after PCI for culprit lesions. Later, the incidence of MACEs related to NSA showed a gradual separation, and by 5 years of follow-up, MACEs in the CKD group were significantly more common than they were in the non-CKD group, which was mainly caused by increased rates of cardiac death and revascularization for NSA.

Based on the subtle 1-year changes in the composition and nature of NSA, as shown by OFR measurement, combined with the 5-year clinical follow-up data, we speculate that the main difference in the outcome of NSA in patients with mild CKD vs. without CKD is the increase in calcification components, and the change in calcification structure is more characterized by the thickness and luminal protrusion. It is likely that the inflammatory component of NSA (macrophage TAVn/PAV), as measured by OFR, is not significantly different between the CKD group and the non-CKD group but leads to an increase in NSA-related MACEs at the 5-year follow-up in CKD patients. This hypothesis needs to be verified by clinical studies, and whether the NSA calcification of mild CKD patients can be alleviated by anti-inflammatory therapy needs to be confirmed by clinical trials.

## Limitations

1) This was a retrospective study with a small sample, and no patients with advanced CKD (stage 3~5 CKD) were enrolled. 2) Without dynamic monitoring of renal function, it was impossible to confirm whether the deterioration of renal function plays a role in the change in NSA. 3) LLT based on statins in all patients did not meet the criteria recommended by guidelines. 4) There was no record of serum inflammatory markers, so our findings do not explain whether anti-inflammatory therapy can delay the calcification progression of NSA. 5) Because OCT does not clearly distinguish between intimal and media calcification (there is no clear boundary), our study was unable to quantify whether the increase in calcification in patients with mild CKD was primarily in the intimal or media portion at 1-year.

## Conclusions

Despite a comparable reduction in LDL-C levels after LLT, more PP with an increase in the calcium volume of TAVn and a worsening of morphology (more protrusion into the lumen) of NSA at the 1-year follow-up and a higher corresponding incidence of NSA-related MACEs at the 5-year follow-up were observed in mild CKD patients than non-CKD patients.

## Acknowledgments

The authors thank Professor Sheng-xian Tu and his colleagues for providing OFR analysis software (OFR, Pulse Medical Imaging Technology, Shanghai, Co., Ltd.) and professional training.

## Author Contributions

**Conceptualization:** De-lu Yin, Lang Li, Fei Ye.

**Data curation:** Jia-cong Nong, Wei You, Yi-fei Wang, Xiang-qi Wu, Zhi-ming Wu, Xiao-han Kong, Hai-bo Jia.

**Formal analysis:** Jia-cong Nong, Yi-fei Wang, Xiang-qi Wu, Zhi-ming Wu, Xiao-han Kong, Hai-bo Jia.

**Investigation:** Wei You, Yi-fei Wang, Yi Xu, Tian Xu, Pei-na Meng.

**Methodology:** De-lu Yin, Lang Li, Fei Ye.

**Project administration:** De-lu Yin, Lang Li, Fei Ye.

**Supervision:** Fei Ye.

**Validation:** Jia-cong Nong, Yi Xu, Tian Xu, Pei-na Meng, De-lu Yin, Fei Ye.

**Visualization:** Wei You, Yi-fei Wang.

**Writing – original draft:** Jia-cong Nong, Yi-fei Wang, Lang Li.

**Writing – review & editing:** Lang Li, Fei Ye.

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
