## [Decision Letter · Decision Letter 0]

28 Dec 2023

PONE-D-23-37592Dynamic natural components and morphological changes in nonculprit subclinical atherosclerosis in patients with acute coronary syndrome and mild chronic kidney disease at the 1-year follow-up and clinical significance at the 5-year follow-upPLOS ONE

Dear Dr. 叶,

Thank you for submitting your manuscript to PLOS ONE. After careful consideration, we feel that it has merit but does not fully meet PLOS ONE’s publication criteria as it currently stands. Therefore, we invite you to submit a revised version of the manuscript that addresses the points raised during the review process.

We look forward to receiving your revised manuscript.

Kind regards,

Hean Teik Ong

Academic Editor

PLOS ONE

Journal Requirements:

[None].

6. We note that Figure 2 and 3 in your submission contain copyrighted images. All PLOS content is published under the Creative Commons Attribution License (CC BY 4.0), which means that the manuscript, images, and Supporting Information files will be freely available online, and any third party is permitted to access, download, copy, distribute, and use these materials in any way, even commercially, with proper attribution. For more information, see our copyright guidelines: http://journals.plos.org/plosone/s/licenses-and-copyright.

a. You may seek permission from the original copyright holder of Figure 2 and 3 to publish the content specifically under the CC BY 4.0 license.

7. Please include your tables as part of your main manuscript and remove the individual files. Please note that supplementary tables (should remain/ be uploaded) as separate ""supporting information"" files".

Reviewers' comments:

Reviewer's Responses to Questions

**Comments to the Author**

1. Is the manuscript technically sound, and do the data support the conclusions?

Reviewer #1: Yes

Reviewer #2: Yes

2. Has the statistical analysis been performed appropriately and rigorously? 

Reviewer #1: Yes

Reviewer #2: Yes

3. Have the authors made all data underlying the findings in their manuscript fully available?

Reviewer #1: Yes

Reviewer #2: Yes

4. Is the manuscript presented in an intelligible fashion and written in standard English?

Reviewer #1: Yes

Reviewer #2: Yes

5. Review Comments to the Author

Reviewer #1: 1. Classification of CKD should use the CKD-Epi 2021 equation rather than the MDRD equation

2. Has Tables 1-3 been submitted? Does not seem to be included in the manuscript.

3. The change in renal function described on page 17 from line 9 to 16 is confusing and should be summarised

4. Was there an increase in calcification in the media?

5. There is a lack of correlation between lesions detected and clinica outcomes

Reviewer #2: Need to rephrase these sentences in Page 22

“ Although many factors are associated with coronary artery calcification progression,

8 such as old age, smoking, diabetes and CKD[CKD [42,43], the current theory is that intimal

9 calcification is related to atherosclerosis, and inflammation contributes, while the

10 characteristics of coronary artery calcification in patients with CKD (especially stages 4-5)

11 are characterized by calcification of the medial layer[39,40,44]. However, atherosclerotic

12 processes dominate in patients with early stages of CKD (stages 1-2)[2) [45,46]. The actual

13 situation is that in our daily practice, the intima and media of the coronary artery are both

14 calcified in CKD patients, which may be related to the participation of multiple factors[34]. ”

6. PLOS authors have the option to publish the peer review history of their article (what does this mean?). If published, this will include your full peer review and any attached files.

Reviewer #1: No

Reviewer #2: **Yes: **Lee Boon Chye

---

## [Author Response · Author response to Decision Letter 0]

15 Mar 2024

We have revised our original manuscript one by one according to the opinions of editors and reviewers, and uploaded our files. Please review them again, thank you.

---

## [Editor Report · Decision Letter 1]

8 Apr 2024

Dynamic natural components and morphological changes in nonculprit subclinical atherosclerosis in patients with acute coronary syndrome and mild chronic kidney disease at the 1-year follow-up and clinical significance at the 5-year follow-up

PONE-D-23-37592R1

Dear Dr. 飞 叶,

We’re pleased to inform you that your manuscript has been judged scientifically suitable for publication and will be formally accepted for publication once it meets all outstanding technical requirements.

Kind regards,

Hean Teik Ong

Academic Editor

PLOS ONE
---

## [Editor Report · Acceptance letter]

10 May 2024

PONE-D-23-37592R1 

PLOS ONE

Dear Dr. Ye, 

I'm pleased to inform you that your manuscript has been deemed suitable for publication in PLOS ONE. Congratulations! Your manuscript is now being handed over to our production team.

Kind regards, 

on behalf of

Dr. Hean Teik Ong 

Academic Editor

PLOS ONE